**Subject Category:**
Biology (whole organism)

cognition/evolution/psychology

unsolvable task, Czechoslovakian Wolfdog, Labrador Retriever, German Shepherd, dingoes, breed differences

**Author for correspondence:**
Emanuela Prato-Previde
e-mail: emanuela.pratoprevide@unimi.it

# Wolf-like or dog-like? A comparison of gazing behaviour across three dog breeds tested in their familiar environments

Veronica Maglieri[1], Emanuela Prato-Previde[2],
Erica Tommasi[1] and Elisabetta Palagi[1,3]

[1]Unità di Etologia, Dipartimento di Biologia, Università di Pisa, Via Volta 6, 56126 Pisa, Italy
[2]Dipartimento di Fisiopatologia Medico-Chirurgica e dei Trapianti, Università degli Studi di Milano, Milano, Italy
[3]Museo di Storia Naturale, Università di Pisa, Via Roma 79, Calci, 56011 Pisa, Italy

EP-P, 0000-0001-8425-1939; EP, 0000-0002-2038-4596

Human-directed gazing, a keystone in dog–human communication, has been suggested to derive from both domestication and breed selection. The influence of genetic similarity to wolves and selective pressures on human-directed gazing is still under debate. Here, we used the 'unsolvable task' to compare Czechoslovakian Wolfdogs (CWDs, a close-to-wolf breed), German Shepherd Dogs (GSDs) and Labrador Retrievers (LRs). In the 'solvable task', all dogs learned to obtain the reward; however, differently from GSDs and LRs, CWDs rarely gazed at humans. In the 'unsolvable task', CWDs gazed significantly less towards humans compared to LRs but not to GSDs. Although all dogs were similarly motivated to explore the apparatus, CWDs and GSDs spent a larger amount of time in manipulating it compared to LRs. A clear difference emerged in gazing at the experimenter versus owner. CWDs gazed preferentially towards the experimenter (the unfamiliar subject manipulating the food), GSDs towards their owners and LRs gazed at humans independently from their level of familiarity. In conclusion, it emerges that the artificial selection operated on CWDs produced a breed more similar to ancient breeds (more wolf-like due to a less-intense artificial selection) and not very human-oriented. The next step is to clarify GSDs' behaviour and better understand the genetic role of this breed in shaping CWDs' heterospecific behaviour.

# 1. Introduction

Studies on social cognition have demonstrated that domestic dogs (*Canis familiaris*) communicate in a complex way with humans by understanding and responding to a variety of gestures and communicative signals (e.g. [1–4]). In particular, dogs are able to communicate with humans in a variety of contexts via visual and auditory signals, including gazing and gaze alternation [5–12].

The role of genetic and environmental processes underlying the communicative abilities of dogs is still a matter of debate. Some scholars claim that dogs' abilities in heterospecific communication stem from the domestication process (e.g. [1,6,13]); others underline that they could be a by-product of early selection for tameness [6,14]. There is also evidence suggesting that more recent artificial selection in certain breeds or breed groups probably affected the communicative behaviour of modern dog breeds [15–20]. Finally, developmental and socio-environmental factors seem to be as relevant as genetic ones in shaping dogs' complex abilities to understand and respond to hetero- and conspecific cues [21–23].

Studies investigating the role of genetic and developmental factors on dogs' capacity to respond to human communicative cues (e.g. different types of pointing, head turning, gazing) are rather numerous [3,10,24–27]; on the other hand, those considering how these aspects may have influenced dogs' ability to produce communicative signals towards humans are still more limited [5,6,18,20,28].

In dog communicative interactions, gazing behaviour plays a central role in building, facilitating and maintaining relationships and social bonds both at intra- and interspecific level [6,10,15,29–31]. Thus, a deeper understanding of how artificial selection, developmental factors and social environment contributed to modern dog's gazing behaviour is relevant both at theoretical and practical levels.

Within dog–human communication, the classical approach to the investigation of gazing behaviour is to create a social situation in which it is impossible for the animal to get a resource which is located out of his/her reach, i.e. the 'unsolvable task'. In this task, after a number of successes in obtaining a reward by manipulating an apparatus, the subject is confronted with an unsolvable version of the task, during which the subject may engage in visual communication with humans to obtain the resource (see [10] for an extensive review).

This experimental approach has been used to compare gazing behaviour in dogs [18,20,32] and other canids, i.e. wolves and dingoes [6,33–36]. Differently from dogs, hand-reared wild canids (wolves and dingoes) gaze at humans rarely and tend to solve the task independently [6,34,36].

Differences in human-directed gazing are also reported across different breeds and groups of dog breeds. In a study based on breed group comparisons, Passalacqua *et al.* [18] reported that Primitive breeds gazed at humans less than Hunting/Herding breeds selected for 'cooperative' work with humans [16]. The authors also found that such divergences in gazing behaviour emerged along with ontogeny and experience with interspecific social communication. Recently, Konno *et al.* [20] found that a group of ancient breeds (genetically closer to the wild ancestor) gazed less and for a shorter time towards humans compared to more recent groups of breeds. The genetic basis of gazing behaviour is also supported by studies providing evidence for an association between owner-directed gazing behaviour in an unsolvable task and polymorphisms in the dog DRD4 gene [37].

As a whole, these findings suggest that the genetic changes produced by domestication and later by artificial selection played, and are still playing, a role in shaping human-directed gazing behaviour [38].

Gazing behaviour also depends on non-genetic factors. For example, Marshall-Pescini *et al.* [32] demonstrated that different types of training can affect the ways dogs communicate with their owners. In the unsolvable task, agility dogs gazed at humans for a longer time compared to search-and-rescue dogs that, conversely, relied more on auditory cues to communicate with humans. Similarly, D'Aniello *et al.* [39] reported that dogs trained for water rescue gazed more at humans compared to untrained dogs of the same breeds. Recently, D'Aniello & Scandurra [40] showed that Labrador Retrievers (LRs) living in a kennel since birth exhibited a lower level of gazing towards humans compared to pet dogs of the same breed, suggesting that a limited human contact can compromise the ontogeny of this behaviour.

The aim of this study is to further investigate the role of artificial selection in shaping gazing behaviour in dogs. In this view, we selected adult pet dogs belonging to three breeds, namely Czechoslovakian Wolfdogs (CWDs), German Shepherd Dogs (hereinafter German Shepherds (GSs)) and LRs, and tested them in the classical impossible task paradigm, to allow comparisons with previous studies. To control for differences in life experiences and interspecific social communication, all the dogs tested were kept for companionship, lived in the same household with their owner and had no specific training. To a greater extent than in other previous studies, we carefully attempted to match our subjects for potential developmental variables selecting a sample of dogs as homogeneous as possible with regard to the social environment and the relationship with the owner. Indeed, developmental and socio-environmental

factors seem to be as relevant as genetic ones in shaping dogs' communicative abilities and problem-solving abilities [40–42]. Moreover, to minimize external interference during the task, that could be a source of distraction or anxiety in the subjects, the tests were carried out in familiar environments to the dogs. Before testing, the experimenters underwent a socialization phase with each dog and the regular daily activities of the dogs were never modified.

The breeds selected for this study are good models to test the influence of selective pressures on dogs' ability to engage in communication with humans through gazing for a number of reasons.

Previous studies assessing breed differences in gazing with the unsolvable task have focused on breed groups comprising a mixture of breeds, each represented by a limited number of subjects [18,20]: thus, testing specific breeds with larger and homogeneous samples may provide a different and useful approach. Additionally, these previous studies have produced partially different results: Konno et al. [20] found that Ancient dog breeds (i.e. more wolf-like breeds) are less likely to produce spontaneous gaze signals towards humans; Passalacqua et al. [18] reported that both Primitive and Molossoid groups showed similar gazing behaviour but were both outranked by Hunting/Herding breeds.

The GS and the LR are two modern breeds resulting from the domestication process and the subsequent process of artificial selection common to all modern breeds, whereas the CWD is a recent dog breed, created about 60 years ago through a hybridization process (GS and wild Carpatian Wolf) in a quite limited time frame. There is some evidence that these breeds differ in the functional roles for which they were selected as well as in their behavioural characteristics [16,43–46].

In particular, the CWD originated from the hybridization between the GS and the wild Carpatian Wolf in the 1950s. The initial aim of the crossing was to select hard-working dogs with enhanced health, strength and night vision. In 1989, after the official approval of CWD as breed standard by the Fédération Cynologique Internationale (FCI), the artificial selection favoured wolf-like morphological features [43]. Hence, this breed has not been directly selected for the expression of neotenic behavioural traits, such as low aggressiveness and high levels of confidence, as it probably occurred in the early domestication phase, which led from wolves to dogs [44]. Notwithstanding their wolf-like appearance, CWDs lack wolf mtDNA haplotypes, but a lot of males still carry Y-haplotypes of wolf origin and many subjects still carry wolf microsatellites [43–44]. This breed is growing in popularity and is spreading in several countries even as a pet. Interestingly, during recent years (since the 1950s), the human-controlled hybridization between dogs and their wild ancestor gave rise to other new breeds like the CWD: the Sarlos Wolfdog and the Lupo Italiano. Our study could help to understand how these new breeds behave.

The GS and the LR are common modern pure breeds that belong to different breed groups [47,48]. The GS is an interesting model due to its genetic closeness to the CWD and to its widespread use as guard dog [45], while the LR underwent a long process of selection to work close to humans and collaborate with them [16]. Both GS and LR are commonly kept as pets in different countries.

Looking at the literature, it emerges that studies comparing GSs and LRs are limited: a behavioural study carried out by Wilsson & Sundgren [49] on a large sample of GSs and LRs revealed that Retrievers were more cooperative and affable than Shepherds, and a more recent study carried out using the C-BARQ questionnaire outlined some differences in the behaviour of these breeds [45,50]. Finally, in a study aimed at assessing breed differences in the use and learning of interspecific gazing responses to get an out-of-reach piece of food, Jakovcevic and colleagues [51] found differences in gazing behaviour between three popular breeds, namely Retrievers, GSs and Poodles, and reported that Retrievers gazed for longer compared to the other two groups to obtain the food, even in the absence of a previous training phase.

Therefore, while there is some evidence on interspecific gazing behaviour in GSs and LRs [39–40,51], to our knowledge, this is the first study on human-directed gazing behaviour in CWDs and the first attempt, since the official approval of the breed by the FCI, to provide scientific evidence on behavioural/communicative characteristics of this breed.

If artificial selection and dogs' 'type of work' influence gazing behaviour towards humans [16,51], we expect to find a gradient in the expression of this behaviour in the three breeds, with CWDs showing a low level of gazing, LR exhibiting a high level of gazing and GS an intermediate one. Moreover, the CWD might have a dog-wolf hybrid behaviour similar to that observed in dingoes [34].

Since there is evidence that wolves are more independent and 'persistent' than dogs when faced with a problem-solving task, either possible or impossible, and less prone than dogs to give up [6,33,36,52], we also expect CWDs to spend more time manipulating the apparatus than LRs and GSs.

Finally, some studies have reported that dogs preferentially gazed at the owner compared to an unfamiliar experimenter who had manipulated the food during the task [7]; whereas others did not

reveal any bias in gazing behaviour [39]. For this reason, here we also evaluated possible breed differences in the preference for gazing towards the owner (the person from whom the animal receives support regularly and with whom shares a strong attachment, [53]) or the experimenter (the person manipulating the food in the apparatus). If gazing is a communicative behaviour mainly driven by the relationship with the owner, the dogs should gaze preferentially at the owner (relational response); on the other hand, if gazing is driven by the opportunistic motivation to reach the food, dogs should gaze more at the experimenter (opportunistic response). Owing to its recent breed origin and the type of selection, we expect that the CWD would be more prone to specifically gaze at the experimenter compared to the owner (opportunistic response). Conversely, we expect that the gazing response of LRs and GSs would be more driven by their relationship with owners (relational response).

# 2. Material and methods

## 2.1. Subjects

All the dogs were recruited through personal contacts, advertisements on the Internet, in parks and at veterinary clinics and by word of mouth in the Provinces of Livorno, Pisa, Firenze and Lucca (Tuscany, Italy). A total of 23 CWDs (11 females; 12 males), 18 LRs (10 females; 8 males) and 15 GSs (9 females; 6 males) were tested. All the dogs were kept for companionship, lived within the human household, were used to meet people and had either no or only basic training experience. In order to have a sample of subjects as homogeneous as possible with regard to the social environment and the relationship with the owner, we selected dogs that had been acquired at the same age (i.e. two-months-old) and who spent most of their time with the owner (i.e. also sleeping with him/her).

## 2.2. Training of the experimenters

During a preliminary training, the two experimenters involved in the data collection were asked to simulate the actions and movements they would perform during the test trials. This aimed at obtaining a high level of behavioural homogeneity between experimenters during the test. During the simulation, a female Border Collie was tested in both possible and impossible tasks. This subject was obviously not included in the analysis.

## 2.3. Apparatus and the procedure

The apparatus consisted of a 15 cm metal strainer placed upside down over a few titbits of food on a 90 × 60 cm plywood. The strainer could be either moved off the platform or overturned to obtain the food (possible condition or task) or it could be securely screwed to the plywood so the food could not be accessed (impossible condition or task).

All the tests were conducted in a place familiar to the dogs, i.e. their garden or a park they regularly frequented), according to the owner's availability. Owners were asked not to feed their dogs for at least 4 h prior to testing.

Before the test, the dogs were allowed to familiarize with the experimenters. During this familiarization period, which could last from 10 to 15 min, the owners were thoroughly instructed on how to behave during the test, to make sure they would behave appropriately without making mistakes. We also explained to the owners the aim of the study in order to give them basic information to better understand the rationale of the test and to make them involved in the project.

The test consisted of six 'solvable' trials (possible task) in which dogs could obtain the food by manipulating the metal container, followed by an 'unsolvable' trial (impossible task) in which the container was fixed onto the plywood. In all trials, the owner and the researcher maintained the same position, i.e. at either side and two steps back—40/50 cm—from the plywood on which the container was placed. During the entire test period, the owner and researcher looked straight ahead and ignored the dog (i.e. neither speaking, looking or touching it). In the 'solvable' trials, dogs were positioned between (or just in front of) the owner's legs and held by collars. The dog witnessed the experimenter squatting down and placing one piece of food under the metal strainer. The food consisted in a small piece of Wudy Aia® chicken Vienna sausage or a small piece of food provided by the owner in the case of dogs' special needs.

As soon as the experimenter reached her position, the owner was instructed to leave the dog who was then allowed to move freely around the apparatus and within the testing area. During the trials, the dogs

were never kept on a leash. Each trial was interrupted after a maximum of 1 min or as soon as the dog obtained the food. Only the dogs that succeeded in at least four trials in obtaining the food in the 'solvable' task were admitted to the 'unsolvable' one. Trials were presented in sequence with no interruption.

In the 'unsolvable' task, an identical metal strainer was fixed to the plywood with the same quantity of food inside. The experimenter placed the plywood on the ground in front of the dog (held by its owner) and then squatted down next to the container, placing a hand over it and with the other hand pretending to place food under it. This was done to make the actions on the container as similar as possible to those carried out in the 'solvable' task. In line with many previous studies on gazing towards humans in pet dogs [9,18,20,32,39] and based on the evidence that dogs are quick to seek human contact, especially eye contact, when faced with a problem, the 'unsolvable' task lasted 1 min. All trials were video-recorded using a video camera (Canon EOS® 1100D) positioned on a tripod located in the testing area.

An additional, mobile camera (Sony HANDYCAM® DCR-SR32E) was used by the second experimenter if the dog moved out of the view field of the fixed camera. The videos permitted an accurate analysis of the behaviour of the dogs towards the metal strainer, the owner and the experimenter.

## 2.4. Video analysis

The program VLC® media player with the 'Jump to time' extensions was used to analyse videos and calculate the exact duration of each behavioural item. Dogs' behaviours were coded during both the 'solvable' and 'unsolvable' tasks,

We clustered the behaviours of dogs according to the following categories: (i) any behaviour performed by the dog to olfactorily and visually explore the apparatus without any physical contact with the apparatus (e.g. sniffing, looking at the metallic strain); (ii) any behaviour to manipulate the apparatus (e.g. pawing, licking, scratching, biting, nibbling, pulling and pushing the metallic strain); (iii) any 'gazing' event at the owner or at the experimenter (the dog gazes at the person by turning or lifting its head); (iv) any 'looking away' event from the apparatus and humans or going away from the experimental setting; (v) under the category of 'any behaviour' towards humans (e.g. rubbing, nosing, licking pawing a hand/leg or jumping up). For each behaviour, we recorded the duration (with a 0.2 s accuracy). For gazing behaviour, we also recorded the number of times the dog gazed towards humans, the latency of gazing (the time elapsing between the start of the trial and the first gaze) and the frequency of gaze alternation behaviour, i.e. two-way gaze alternation between person and container within 2 s and vice versa [18,32].

Before starting the analysis, each experimenter underwent a training period to learn to analyse videos frame-by-frame. A random selection of videos (about 20%) was used to evaluate inter-observer reliability [54] during all the course of the analysis at regular intervals. Moreover, another random selection of videos (about 20%) was also used to evaluate intra-observer reliability. The same observer analysed the same videos at least twice and checked for the scores and the duration of each behavioural item. For both inter-observer reliability and intra-observer reliability, the Cohen's kappa was never below 0.85.

## 2.5. Comparisons with a previous study on dingoes

To compare dingoes with the dog breeds tested in our study, we extracted the data from Smith & Litchfield ([34], table 2, p. 6). In this paper, the authors examined gazing towards humans in these wild canids using the same apparatus devised by Miklósi *et al.* [6] to test wolves. Even though their apparatus differed from the one we used in the current study, the task was a simple manipulation task (i.e. a bin-opening and a rope-pulling task) that comprised a number of solvable trials and then an unsolvable trial. So conceptually it was similar to our task and allowed to analyse looking behaviour towards humans and problem-solving strategies. The authors reported both the latency and the time spent by dingoes in looking at the human ('looking back' at the caregiver/experimenter). As the impossible task carried out by Smith & Litchfield [34] lasted 120 s, to make their data comparable with ours, we considered only the first 60 s reported in table 2 (p. 6).

## 2.6. Statistical analysis

When the data were normally distributed (Kolmogorov–Smirnov, $p > 0.05$), parametric statistics were used; in all the other cases, we used non-parametric statistics. The behaviour of the three breeds was compared the exact Kruskal–Wallis test (corrected for group number). Intra-group analyses were carried out by using the binomial test, the exact Wilcoxon-signed ranks test and the paired sample

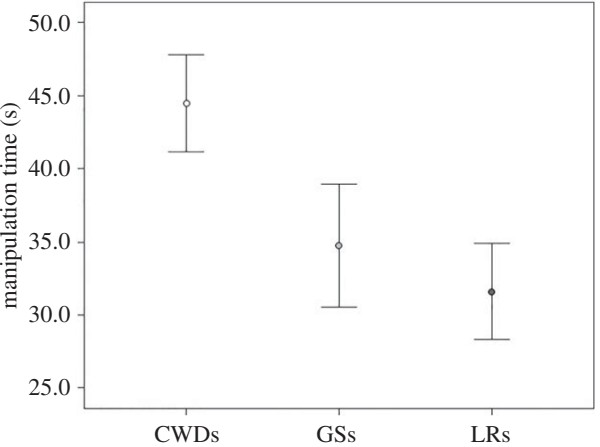

**Figure 1.** Mean ± s.e. of manipulation time (s) during the 'impossible task' in the three groups of breeds (CWDs, Czechoslovakian Wolfdogs; GSs, German Shepherds; LRs, Labrador Retrievers).

*t*-test (in the case of the normal distribution of the data). The Friedman test was used to compare the dog's performance across the six 'solvable trials'. The effect size was estimated using the non-parametric Rosenthal's *r* coefficient. All the analyses were carried out via SPSS® 20.0 software. All the analyses were two-tailed with the alpha value set at 0.05.

# 3. Results

Out of the 56 dogs tested, 43 (77%) solved the possible task at least four times thus gaining access to the unsolvable trial. The subjects which passed the first step included 17 CWDs (8 females, 9 males; mean age = $36.59 \pm 5.60$ s.e.; $age_{min} = 5$ months; $age_{max} = 84$ months), 14 LRs (8 females, 6 males; mean age = $49.36 \pm 7.82$ s.e.; $age_{min} = 5$ months; $age_{max} = 96$ months) and 12 GSs (8 females and 4 males, mean age = $41.25 \pm 6.23$ s.e.; $age_{min} = 9$ months; $age_{max} = 72$ months). No significant age differences emerged across the three groups (Kruskal–Wallis test: $\chi^2 = 1.76$; $N_{CWD} = 17$; $N_{LR} = 14$; $N_{GS} = 12$; d.f. = 2; $p = 0.415$).

## 3.1. Possible (or solvable) task

In each group, the latency to success significantly decreased across the six possible trials (CWD—Friedman test, $\chi^2 = 19.90$, $N = 17$, d.f. = 5, $p = 0.001$; Dunnett *post hoc* test: $trial_1$ versus $trial_6$ $q = 14.85$, $p < 0.01$; LR—$\chi^2 = 17.76$, $N = 12$, d.f. = 5, $p = 0.003$; Dunnett *post hoc* test: $trial_1$ versus $trial_6$ $q = 4.61$, $p < 0.01$; GS—$\chi^2 = 11.16$, $N = 7$, d.f. = 5, $p = 0.039$; Dunnett *post hoc* test: $trial_1$ versus $trial_6$ $q = 4.97$, $p < 0.01$). This analysis included only those dogs who passed all the six solvable trials.

Considering all the subjects admitted to the 'unsolvable' task, no breed difference emerged in the latency to success during the first trial in which the subject reached the reward (Kruskal–Wallis $\chi^2 = 2.849$; $N_{TOT} = 43$; $N_{CWD} = 17$; $N_{GS} = 12$; $N_{LR} = 14$; d.f. = 2; $p = 0.241$).

During the solvable task, only 3 out of 17 CWDs (23.5%) gazed towards humans (binomial test: $N = 17$; $p = 0.018$), while 8 out of 14 LRs (57.14%) and 5 out of 12 GSs (41.6%) showed this behaviour (binomial test: LRs, $N = 14$, $p = 0.209$; GSs, $N = 12$, $p = 0.193$).

During the first possible trial, no differences were found in the preference of gazing (i.e. gazing at the owner versus the experimenter) in CWDs (Wilcoxon-signed ranks test: $T = 0.00$; ties = 14; $N = 17$; $p = 0.250$; $mean_{owner} = 0.00 \pm 0.00$ s.e.; $mean_{experimenter} = 0.12 \pm 0.9$ s.e.), GSs ($T = 0.00$; ties = 8; $N = 12$; $p = 0.125$; $mean_{owner} = 0.03 \pm 0.03$ s.e.; $mean_{experimenter} = 0.37 \pm 0.26$ s.e.) and LRs ($T = 0.00$; ties = 10; $N = 14$; $p = 0.125$; $mean_{owner} = 0.76 \pm 0.76$ s.e.; $mean_{experimenter} = 1.76 \pm 1.57$ s.e.).

## 3.2. Impossible (or unsolvable) task

In the unsolvable task, there was no difference across the three breeds in the time spent in visual (Kruskal–Wallis test: $\chi^2 = 3.46$; $N_{CWD} = 17$, $N_{LR} = 14$, $N_{GS} = 12$; d.f. = 2; $p = 0.177$) and olfactory exploration of the apparatus ($\chi^2 = 4.99$; $N_{CWD} = 17$, $N_{LR} = 14$, $N_{GS} = 12$; d.f. = 2; $p = 0.083$). On the contrary, there was a significant difference in the amount of time spent manipulating the apparatus (Kruskal–Wallis test: $\chi^2 = 6.83$, $N_{CWD} = 17$; $N_{LR} = 14$; $N_{GS} = 12$; d.f. = 2; $p = 0.033$; figure 1). CWDs spent

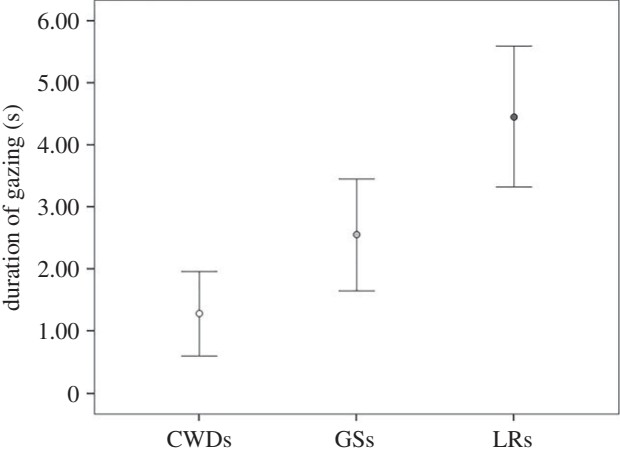

**Figure 2.** Mean ± s.e. of gazing duration (s) during the 'impossible task' in the three groups of breeds (CWDs, Czechoslovakian Wolfdogs; GSs, German Shepherds; LRs, Labrador Retrievers).

more time in manipulating the apparatus compared to LRs (Dunn *post hoc* test: $Q = 2.46$; $p < 0.05$, Rosenthal's $r = 0.43$), but there was no significant difference between CWD and GS ($Q = 1.81$; $p > 0.05$, Rosenthal's $r = 0.29$) and between GS and LR ($Q = 0.47$; $p > 0.05$, Rosenthal's $r = 0.18$).

As regard gazing behaviour, 9 out of 17 CWDs (53%) (binomial test: $p = 0.185$), 9 out of 12 GSs (75%) (binomial test: $p = 0.054$) and 13 out of 15 LRs (86%) (binomial test: $p = 0.001$) gazed towards humans.

The duration of gazing significantly varied across the three breeds (Kruskal–Wallis $\chi^2 = 8.34$; $N_{CWD} = 17$, $N_{LR} = 14$, $N_{GS} = 12$; d.f. = 2; $p = 0.015$; figure 2). CWDs gazed towards humans for shorter periods of time compared to LRs (Dunn *post hoc* test: $Q = 2.83$; $p < 0.05$, Rosenthal's $r = 0.45$) but there was no difference in the time spent gazing at humans between CWDs and GSs ($Q = 1.50$; $p > 0.05$, Rosenthal's $r = 0.25$) and between LRs and GSs ($Q = 1.14$; $p > 0.05$, Rosenthal's $r = 0.23$). Finally, the three breeds did not differ in the latency of the first gaze towards humans (Kruskal–Wallis $\chi^2 = 0.558$; $N_{CWD} = 8$, $N_{LR} = 13$, $N_{GS} = 9$; d.f. = 2; $p = 0.757$).

### 3.3. Owner versus experimenter

There were significant differences in the dogs' preference for gazing at the owner or at the experimenter in the impossible task. In particular, CWDs preferentially gazed at the experimenter (Wilcoxon-signed ranks test: $T = 3.00$; ties = 9; $N = 17$; $p = 0.036$; mean$_{owner} = 0.44 \pm 0.28$ s.e.; mean$_{experimenter} = 0.85 \pm 0.41$ s.e., Rosenthal's $r = 0.33$). GSs showed a significant preference for gazing towards the owner (paired sample *t*-test = 2.41; d.f. = 11; $p = 0.035$; mean$_{owner} = 1.98 \pm 0.71$ s.e.; mean$_{experimenter} = 0.58 \pm 0.26$ s.e., Rosenthal's $r = 0.33$); no significant differences were found in LRs (paired sample *t*-test = −0.902; d.f. = 13; $p = 0.383$; mean$_{owner} = 1.75 \pm 0.83$ s.e.; mean$_{experimenter} = 2.70 \pm 0.71$ s.e., Rosenthal's $r = 0.19$) (figure 3).

As for the frequency of gaze alternation between the apparatus and the human subject, significant breed differences emerged (Kruskal–Wallis test: $\chi^2 = 8.23$; $N_{CWD} = 17$, $N_{LR} = 14$, $N_{GS} = 12$; d.f. = 2; $p = 0.016$). Gaze alternation was significantly less frequent in CWDs than in LRs (Dunn *post hoc* test: $Q = 2.81$; $p < 0.05$, Rosenthal's $r = 0.374$), but comparable with the frequency obtained in GSs ($Q = 1.06$; n.s., Rosenthal's $r = 0.188$). No significant differences in the frequency of gaze alternation emerged between LRs and GSs ($Q = 1.57$; n.s.; Rosenthal's $r = 0.214$).

### 3.4. Comparisons with a previous study on dingoes

In their study on dingoes, Smith & Litchfield ([34], table 2, p. 6) reported that 7 out of 12 subjects (58%) gazed towards humans during the impossible task (binomial test: $p = 0.193$). Comparing the duration of gazing towards humans across the four groups of subjects (dingoes, Smith & Litchfield [34] and CWDs, GSs, LRs, this study), a significant difference emerged (Kruskal–Wallis, $\chi^2 = 11.51$; $N_{CWD} = 17$, $N_{DINGOES} = 12$, $N_{LR} = 14$, $N_{GS} = 12$; d.f. = 3; $p = 0.009$; figure 4). The amount of time spent gazing at humans by dingoes was comparable to that spent by our CWDs (Dunn *post hoc* test: $Q = 0.025$; $p > 0.05$, Rosenthal's $r = 0.226$) and GSs ($Q = 1.44$; $p > 0.05$, Rosenthal's $r = -0.034$). Conversely, LRs spent a higher amount of time gazing at humans than dingoes (Dunn *post hoc* test: $Q = 2.69$; $p < 0.05$, Rosenthal's $r = 0.293$). Finally, the four

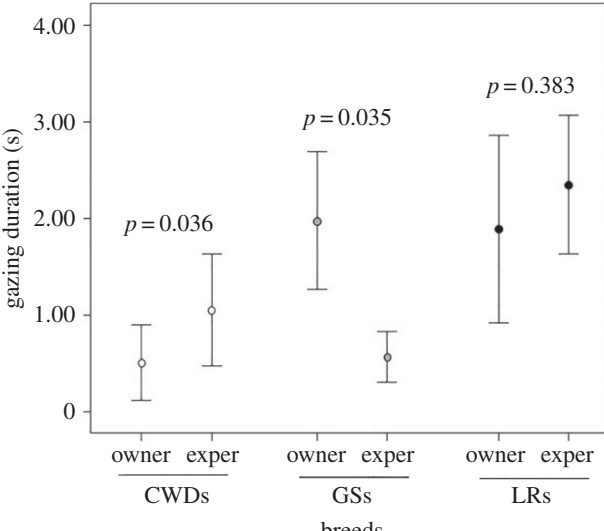

**Figure 3.** Mean ± s.e. of gazing duration (s) towards the owner and the experimenter during the 'impossible task' in the three groups of breeds (CWDs, Czechoslovakian Wolfdogs; GSs, German Shepherds; LRs, Labrador Retrievers).

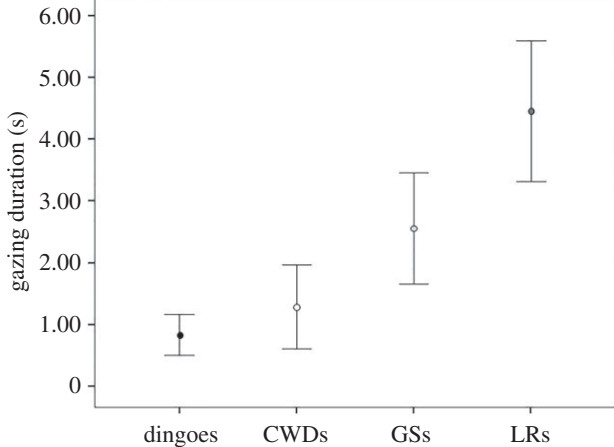

**Figure 4.** Mean ± s.e. of gazing duration (s) during the 'impossible task' in dingoes and in the three groups of breeds (CWDs, Czechoslovakian Wolfdogs; GSs, German Shepherds; LRs, Labrador Retrievers).

groups of subjects did not differ in the latency of the first gaze towards humans (Kruskal–Wallis $\chi^2 = 1.89$; $N_{CWD} = 17$, $N_{DINGOES} = 12$, $N_{LR} = 14$, $N_{GS} = 12$; d.f. = 3; $p = 0.595$).

## 4. Discussion

In the solvable trials, all the dogs quickly learned how to obtain the reward by manipulating the apparatus with their mouths or paws. The ability to solve the task recorded in our groups makes our data comparable with those coming from other studies [6,9,20,32,34,36,40]. No differences emerged in the latency to success during the first trial and the latency to obtain the reward significantly decreased over the six possible trials: this suggests that each group of dogs quickly mastered the task in a similar manner and that all dogs were both physically capable to obtain the reward and sufficiently and equally motivated to do so.

As reported in the study by Marshall-Pescini *et al.* [32], our dogs gazed toward humans to some extent also in the solvable trials, even though with diverse performances across the different groups of breeds. Most CWDs (14 out of 17) never gazed at humans. This result differs from that obtained for GS and LR groups. This finding is in line with that obtained by Miklósi *et al.* [6] on socialized wolves and by Smith & Litchfield [34] on socialized dingoes.

In the unsolvable task, the similar levels of olfactory and visual exploration of the apparatus performed by the three groups of breeds strongly suggest their comparable motivation to reach the reward. However, this similar motivation translated into a different kind of tactics to try and solve the task. The CWD group spent a larger amount of time in physically manipulating the apparatus (high level of persistence in individually solving the task) compared to the LR group; on the other hand, the GS group did not differ either from CWD or LR group, thus classifying as an intermediate form. This finding indicates a strong variability in the level of persistence to manipulate the apparatus in our dog groups despite their similar habits and rearing conditions, with the CWD showing a more 'wolf-like' behaviour, probably due to its genetic closeness to the wild ancestors. Indeed, it has been shown that both juveniles and adult socialized wolves are more persistent (i.e. spend more time in tackling a task) than pet dogs, and dogs in general, in solving different problems [6,33,36,52]. Marshall-Pescini et al. [36] tested in an impossible task four categories of subjects: similarly raised and kept adult wolves and mixed-breed dogs, mixed-breed pet dogs and free-ranging dogs. They found that besides the species and the levels of socialization, wolves were more persistent than all dog groups. In addition, Marshall-Pescini et al. [36] also reported that less persistent animals looked back at the human sooner and longer suggesting a link between a low level of persistence in problem-solving and 'looking back behaviour'. This is an interesting issue that deserves further investigation.

With regard to gazing towards humans, the results obtained via the binomial test for the impossible task are congruent with those obtained in the possible task. CWDs was the only group that did not show the tendency to gaze towards humans, while the other two groups of breeds showed a tendency to look back at humans. These findings are in agreement with the results obtained by Miklósi et al. [6] on hand-reared wolves. Moreover, the comparison between our groups of dog breeds and dingoes via binomial test revealed a substantial similarity between the CWD group and dingoes (data extracted from [34]).

The duration of gazing towards humans significantly differed across the three groups of breeds. The CWD gazed towards humans significantly less than LRs. The duration of gazing towards humans recorded for GSs did not significantly differ from that recorded for either CWDs or LRs.

Considering that all our dogs' rearing and living conditions were similar, as they were all pets living in close contact with their owners (i.e. also sleeping with them), it seems unlikely that the differences in gazing behaviour could be mainly ascribed to the diverse rearing conditions operating on the development of social relationships towards humans. Nevertheless, given that in studies involving pet dogs controlling for all environmental and relational variables is impossible, the existence of some differences in the owners' interaction with CWDs, GSs and LRs cannot be completely excluded. It should be noted, however, that differences in owner–dog interaction may exist not only at the breed level but also at the individual level.

The comparative analysis on gazing duration between dingoes and our groups gave an interesting result. Dingoes, CWDs and GSs did not differ from each other in gazing duration. Our results not only indicate that there is a substantial agreement in gazing response between dingoes and CWDs, but they also highlight the peculiarity of the GS group in performing gazing behaviour. The GS group showed a nuanced behaviour during both solvable and unsolvable tasks that is worth further investigation.

Recently, Sundman et al. [55] reported differences between GS and LR in human-directed social behaviours displayed during a problem-solving task, with Labrador showing more eye contact, and there is evidence that some social skills in dogs have been selected for and further enhanced in some breeds [38,56]. Thus, the nuanced behaviour of GS observed in the current study could depend on some differences in social behaviour between the two breeds, with LR being particularly social, more cooperative and more prone to gaze at humans [51].

In dogs, gaze alternation is a good means to evaluate the capacity to enhance referential communication between the human and an object in the environment [5]. Alterisio et al. [57] suggest that in horses the alternation in gazing could be interpreted by the observer as a request for help. Our data on gaze alternation showed strong similarity with those obtained for the gaze duration, with CWDs performing less gaze alternation compared to LRs but not compared to GSs.

A clear difference between the GS and the CWD emerged in the direction of gazing. While CWDs directed their gazing preferentially towards the experimenter (the only human subject manipulating the food), GSs mainly gazed towards their owners. The preference of the CWDs towards the experimenter suggests that the gazing behaviour could be an opportunistic strategy in this dog. Another possibility could be that CWDs due to their crossing with wolves were more alerted by the presence of an unfamiliar person: however, in the first possible trial, the three breeds did not show

any difference in gazing at the experimenter and thus it seems unlikely that their gazing preference was due to differential alertness and wariness.

On the contrary, the GS gazing response seems to be driven by the relationship shared with their owner. Interestingly, the LR group did not show any preference in experimenter versus owner gazing. They tried to obtain help from humans independently from their level of familiarity. Despite its genetic closeness to the CWD [56–57], the GS (the breed crossed with Carpatian Wolf to obtain the CWD) resulted to be more owner-oriented compared to the CWD. The difference in the preference towards the owner persists also when GSs are compared with LRs. This last finding could be due to the high level of sociability of Labradors [46] that, combined with their high propensity to food [58,59], drive them to turn to humans independently from familiarity. This could be a good issue to explore by enlarging the sample size.

The difference in gazing behaviour between CWDs and LRs and the intermediate position of the GSs in gazing towards humans strongly supports the hypothesis according to which gazing behaviour can have genetic bases [6,18,20,34,37].

In conclusion, our findings suggest that despite the numerous crossing with GSs after the first hybrid litter, the artificial selection operated on CWDs has produced a breed more similar to ancient breeds (more genetically close to wolves due to a less-intense artificial selection) and not particularly human-oriented. However, current results should be interpreted with some care, due to some limitations of the study: although our sample was selected with care, the possibility that, depending on the breed, the owners behaved differently in everyday situations thus influencing the dog behaviour cannot be excluded. Moreover, the sample size was rather limited, also due to the difficulty of recruiting dogs with all the required characteristics (in particular CWDs and GSs) and owners willing to participate. Future comparisons testing a larger sample of dogs using different communicational paradigms as well as problem-solving contexts may lead to a clearer interpretation of the current findings. It would be interesting to clarify the behaviour of GSs in order to better understand the genetic role that this breed had played in shaping the CWD heterospecific behaviour.

Ethics. All procedures were performed in full accordance with Italian legal regulations and the guidelines for the treatments of animals in behavioural research and teaching of the Association for the Study of Animal Behaviour (ASAB). In Italy, observational studies of animal behaviour are considered as procedures not subjected to the National Directive n. 26/14 (transposition of the 2010/63/UE directive on the protection of animals used for scientific purposes, article 1, comma 5), and for them further ethical approval is not requested. Hence, no special permission was needed to carry out this study. All dog owners voluntarily participated, they received an in-depth description of the study and its rationale and their permission to use video-record and to use data in an anonymous form was obtained prior to testing.

Data accessibility. The datasets supporting this article are available from Dryad Digital Repository: https://doi.org/10.5061/dryad.5q168t7 [60].

Authors' contributions. Conception and design (E.P. and E.P.-P.), acquisition of data (V.M. and E.T.), analysis and interpretation of data (V.M., E.P., E.P.-P.); drafting the article (V.M., E.P., E.P.-P.); final approval of the version to be published (V.M., E.T., E.P.-P., E.P.); agreement to be accountable for all aspects of the work in ensuring that questions related to the accuracy or integrity of any part of the work are appropriately investigated and resolved (V.M., E.T., E.P.-P., E.P.).

Competing interests. We have no competing interests.

Funding. This study was self-funded by the authors and their institutions.

Acknowledgements. We wish to thank the owners and the dogs who participated in the study, Alessandro Massolo for useful suggestions and W. Banfi for enlightening discussions.

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
