## [Reviewer comments · Royal Society Open Science]

Review History

RSOS-190946.R0 (Original submission)

Review form: Reviewer 1

Is the manuscript scientifically sound in its present form?

Yes

Are the interpretations and conclusions justified by the results?

Yes

Is the language acceptable?

Yes

Do you have any ethical concerns with this paper?

No

Have you any concerns about statistical analyses in this paper?

Yes

Recommendation?

Accept with minor revision (please list in comments)

Comments to the Author(s)

In their paper "Wolf-Like or Dog-Like?...", the authors carefully assess problem solving responses of genetically distinct dog breeds with similar rearing history to determine whether reliance on human assistance in problem solving--operationalized here as eye gaze during an unsolvable task--has a clear genetic component.

The authors gave Czechoslovakian wolf dogs (a recently create dog/wolf hybrid), German shepherds, and Labrador retrievers an unsolvable task involving food placed beneath a cover bolted to an apparatus. All dogs were companion animals with broadly similar history and no explicit training. Each dog had one minute of exposure to the impossible task, during which their time attempting to manipulate the apparatus and their time gazing at an experimenter and owner were coded.

The authors found that CWDs spent more time working on the apparatus and less time gazing at humans than LRs, with GSs in the middle on these measures. From this they argued that CWDs, a recent wolf hybrid, show less reliance on humans in problem solving, and, given similar rearing for all subjects, this is likely due to genetic differences between established domestic breeds and wolf hybrids (presumably due to contribution of wolf genome/transcriptome).

I believe the authors' methods were generally sound. I especially commend them for attempting to match their subjects for prior rearing history. Many different breeds are likely to have had very different individual histories with explicit training programs that would be likely to affect problem solving and gazing behavior. Many prior studies have not fully accounted for these potential developmental effects.

The paper is well and clearly written and, I think, makes a meaningful contribution to ongoing debate about the genetics underlying dog/human interaction and cooperation.

I have one primary concern with the study and how it was interpreted. Despite some prior work allowing subjects 2 or more minutes for exploring the impossible task, in the current study the subjects had only one minute. I think this limits the confidence with which the data might be interpreted.

The authors note that CWDs spend less time looking at humans than do LRs, and suggest this is because LRs are more reliant on humans. However, from figure 4, it is clear that CWDs spent the majority of the minute actively focused on interacting with the apparatus, leaving much less remaining time for looking at humans/asking for/expecting help.

An alternative interpretation that the authors' data does not clarify is that, instead of CWDs having less interest in humans than LRs, they are just more PERSISTENT in problem solving. It may be that, when one wears out one's self-motivated attempts to solve the problem, each breed might then be equally likely to look to a human for intervention. But it's not clear that CWDs had enough time to meaningfully work beyond their own self-efficacy.

I suspect that self-efficacy is indeed anti-correlated w/ communication w/ humans/supplication to/reliance on humans across a range of contexts. These are related phenomena. But I believe these phenomena could still be disentangled with a more careful study design.

A similar concern -- the authors note that CWDs spend more time looking at experimenters than owners as opposed to LRs. But, again, I wonder about the time frame/amount of time available for looking after finishing self-driven work on the task. Might LRs and GSs have spent more time earlier in the trial looking at experimenters before switching gaze deferentially toward owners? If so, again, we might posit that the CWDs didn't have enough time, post self-driven efforts, to work through the human supplication decision tree.

I don't believe this concern is insurmountable. I do, however, believe the authors need to clearly address possible alternative interpretations to their data given the limitation of time in the current study.

Much more minor concerns:

The general rearing situation of these dogs is similar, which is good. Might it be the case that people interact w/ CWDs differently, even in a home rearing situation? This might at least be addressed. The authors have gone a long way to control for developmental effects, but they have not fully limited all potential avenues.

On line 155, the authors suggest that gazing at the owner might suggest a more "emotional" versus "operational" interest in a human. However, might it not also suggest that the animal has a long prior facultative history with their owner whereby the owner solves lots of problems for the animal? One might just look at one's owner because one's owner usually intervenes in difficult situations, regardless of one's emotional state. In addition, I think it's somewhat loose to suggest that looking at owner then back at task is "asking for help." Certainly we might question whether it bears some similarity to asking for help, but it's much more parsimonious, without specific evidence, to just suggest that the animal is dividing attention between the reward and the means by which prior rewards have often been delivered.

More specific detail on the subjects in the methods/materials would be welcome. Again, I take it the most meaningful contribution here is evidence for genetic, as opposed to developmental, influence on gazing behavior. More detail on life history would be informative.

Line 174 - I did not understand in this training context what was meant by "complete behavioural agreement between experimenters."

Line 334 -- I don't understand this sentence -- how can Marshall-Pescini et al comment on the current (as of yet unpublished) work in an already published work?

385 - The authors suggest that CWD's gazing at experimenters is opportunistic, but might it not also suggest wariness/alertness to a stranger? The familiarization period was short, and in my experience w/ wolves they are VERY attentive to novel humans, in part due to higher arousal/threat sensitivity.

Again, I believe the paper is strong and well written. If the authors can better couch their interpretation in terms of a range of possible interpretations and the study's inherent limitations than I think it certainly merits publication in RSOS.

Review form: Reviewer 2

Is the manuscript scientifically sound in its present form?

Yes

Are the interpretations and conclusions justified by the results?

No

Is the language acceptable?

Yes

Do you have any ethical concerns with this paper?

No

Have you any concerns about statistical analyses in this paper?

Yes

Recommendation?

Major revision is needed (please make suggestions in comments)

Comments to the Author(s)

Review on the manuscript RSOS-190946

„WOLF-LIKE OR DOG-LIKE? A COMPARISON OF GAZING BEHAVIOUR ACROSS THREE DOG PURE BREEDS TESTED IN THEIR FAMILIAR ENVIRONMENTS“

written by Maglieri et al.

Overall opinion

This is a very well-written manuscript, scholarly accurate and easy to follow. My main concern is about the (by my opinion) minimal novelty of the conducted research. There is already a number of articles that compared dogs vs wolves, different dog breeds (including Dingoes) gazing behavior in the unsolvable task. This present manuscript adds a new dog breed, the Czechoslovakian Wolfdog to the picture, but the Authors did not provide a good enough reasoning, why did they compare their behavior to the German Shepherd Dogs and Labrador retrievers. If we consider the CWD as a dog-wolf hybrid (still, the extent of wolf alleles should be specified then), it would be a better approach to compare their behavior to socialized wolves and the German Shepherd Dog – because then we would had the results from the two original parent species and their hybrid.

As the German Shepherd Dogs showed intermediate results (not differing from the CWDs neither from the Labrador Retrievers), it is very hard to argue that the CWDs show wolf-like behavior.

Detailed comments

Title – would be more correct probably to write “...across three purebred dog breeds...”

All along the text the correct name is German Shepherd Dog (GSD) and not “German Shepherd”.

Introduction/rationale – the reasoning about the novelty of this research is not convincing enough. As Authors mention, there are published results about the differences among dog breeds' gazing behavior (Jakovcevic et al., Konno et al., Passalacqua et al.), basically reporting that more ancient/ or independently 'working' dog breeds show less human-directed gazing behavior than the cooperative/ or more 'modern' dog breeds. This manuscript is mainly different from only that aspect that Authors tested Czechoslovakian Wolfdogs (CWD) as one of the three tested dog breeds. Although it is known that this dog breed was originally created about 60 years ago by hybridizing wolves and German Shepherd Dogs, Authors did not provide additional information about the still remaining contribution of 'wolf alleles' to the genotype of current CWDs. As from the earlier papers we know how socialized wolves (Miklósi et al., 2003) and dingoes (Smith & Litchfield, 2013) behave in the “gaze during an unsolvable task” paradigm, in

case of the present manuscript it is not clear, whether the addition of another dog breed with an unquantified amount of wolf influence would create a significant amount of scientific interest towards the results.

Methods – the comparison between the currently collected and earlier dingo data (retrieved from Smith & Litchfield, 2013) is problematic somewhat, because in the earlier study a very different method was used as ‘unsolvable task’.

Results – Lines 276-278: It seems like 0.5 was the expected ratio of subjects that look back at the humans during the solvable task (Binomial test). Why? In case of the solvable task, based on the theory that human-directed gazing is induced by the fact that the dog finds a formerly solvable task as unsolvable, one could expect that no dogs will look back to the human.

Authors should compare of the ratio of dogs gazing at the human and the other gazing-related parameters BETWEEN the two test conditions (solvable vs. unsolvable task), within each dog breeds.

Line 307 – there was no definition for ‘gaze alternation’ in the Methods. What is the exact description of this behavior?

Discussion

Lines 342-357 – Authors argue that Czechoslovakian Wolfdogs show more wolf-like behavior, because their response in the unsolvable task was different from the retrievers’ (less frequent gazing at the humans, more active manipulation of the apparatus). However, it should not be ignored that there was no statistical difference between the performance of CWDs and German Shepherd Dogs in any of the behavioral parameters (as there was no significant difference between the GSDs and the retrievers either). One could say that it is not the CWDs but the retrievers who show the most dog-like behavior... What is with the GSDs? Now are they wolf or dog-like?

Lines 358-361 – Strictly speaking, Binomial test found only a trend ($P=0.054$) in case of German Shepherd Dogs during the unsolvable task, regarding the proportion of dogs that gazed at the human. Therefore

Line 360 – the interpretation of gazing back at the human as “asking for help” does not have a strong basis, it is rather just a commonly (although probably incorrectly) used term.

Line 390 – correctly it should be “Carpathian Wolf” instead of “Carpathian Wolfdog”

Lines 401-403 – the conclusions drawn by the Authors (“In conclusion, despite the numerous crossing with German Shepherds after the first hybrid litter, it seems that the artificial selection operated on Czechoslovakian Wolfdogs has not yet produced a breed showing complete dog-like behavioural features”) is questionable because they did not detect any significant difference between the human-directed gazing behavior of the Czechoslovakian Wolfdog and the German Shepherd Dog.

Decision letter (RSOS-190946.R0)

03-Jul-2019

Dear Dr Prato-Previde,

The editors assigned to your paper ("WOLF-LIKE OR DOG-LIKE? A COMPARISON OF GAZING BEHAVIOUR ACROSS THREE DOG PURE BREEDS TESTED IN THEIR FAMILIAR ENVIRONMENTS") have now received comments from reviewers. We would like you to revise your paper in accordance with the referee and Associate Editor suggestions which can be found

below (not including confidential reports to the Editor). Please note this decision does not guarantee eventual acceptance.

Please submit a copy of your revised paper before 26-Jul-2019. Please note that the revision deadline will expire at 00.00am on this date. If we do not hear from you within this time then it will be assumed that the paper has been withdrawn. In exceptional circumstances, extensions may be possible if agreed with the Editorial Office in advance. We do not allow multiple rounds of revision so we urge you to make every effort to fully address all of the comments at this stage. If deemed necessary by the Editors, your manuscript will be sent back to one or more of the original reviewers for assessment. If the original reviewers are not available, we may invite new reviewers.

- Data accessibility

<http://datadryad.org/submit?journalID=RSOS&manu=RSOS-190946>

- Competing interests

- Authors' contributions

- Acknowledgements

- Funding statement

on behalf of Dr Rosalind Arden (Associate Editor) and Kevin Padian (Subject Editor)
openscience@royalsociety.org

Associate Editor's comments (Dr Rosalind Arden):

Both Reviewers commented very positively on this manuscript. They both consider it well written and an interesting topic.

They each set out some concerns, and some suggestions for enhancing the work.

In addition to the Reviewers comments, I should be grateful if you would mention whether or not your results were corrected for multiple testing, and if so how this was done. A sentence or two about any effect sizes would also help the reader to interpret your results.

Subject Editor Comments to Author:

Thanks for your submission. We have opted for "major revision" mainly because it gives you a bit more time. We look forward to your revised MS.

Comments to Author:

Reviewers' Comments to Author:

Reviewer: 1

Comments to the Author(s)

In their paper "Wolf-Like or Dog-Like?...", the authors carefully assess problem solving responses of genetically distinct dog breeds with similar rearing history to determine whether reliance on human assistance in problem solving--operationalized here as eye gaze during an unsolvable task--has a clear genetic component.

The authors gave Czechoslovakian wolf dogs (a recently create dog/wolf hybrid), German shepherds, and Labrador retrievers an unsolvable task involving food placed beneath a cover bolted to an apparatus. All dogs were companion animals with broadly similar history and no explicit training. Each dog had one minute of exposure to the impossible task, during which their time attempting to manipulate the apparatus and their time gazing at an experimenter and owner were coded.

The authors found that CWDs spent more time working on the apparatus and less time gazing at humans than LRs, with GSs in the middle on these measures. From this they argued that CWDs, a recent wolf hybrid, show less reliance on humans in problem solving, and, given similar rearing for all subjects, this is likely due to genetic differences between established domestic breeds and wolf hybrids (presumably due to contribution of wolf genome/transcriptome).

I believe the authors' methods were generally sound. I especially commend them for attempting to match their subjects for prior rearing history. Many different breeds are likely to have had very different individual histories with explicit training programs that would be likely to affect problem solving and gazing behavior. Many prior studies have not fully accounted for these potential developmental effects.

The paper is well and clearly written and, I think, makes a meaningful contribution to ongoing debate about the genetics underlying dog/human interaction and cooperation.

I have one primary concern with the study and how it was interpreted. Despite some prior work allowing subjects 2 or more minutes for exploring the impossible task, in the current study the subjects had only one minute. I think this limits the confidence with which the data might be interpreted.

The authors note that CWDs spend less time looking at humans than do LRs, and suggest this is because LRs are more reliant on humans. However, from figure 4, it is clear that CWDs spent the majority of the minute actively focused on interacting with the apparatus, leaving much less remaining time for looking at humans/asking for/expecting help.

An alternative interpretation that the authors' data does not clarify is that, instead of CWDs having less interest in humans than LRs, they are just more PERSISTENT in problem solving. It may be that, when one wears out one's self-motivated attempts to solve the problem, each breed might then be equally likely to look to a human for intervention. But it's not clear that CWDs had enough time to meaningfully work beyond their own self-efficacy.

I suspect that self-efficacy is indeed anti-correlated w/ communication w/ humans/supplication to/reliance on humans across a range of contexts. These are related phenomena. But I believe these phenomena could still be disentangled with a more careful study design.

A similar concern -- the authors note that CWDs spend more time looking at experimenters than owners as opposed to LRs. But, again, I wonder about the time frame/amount of time available for looking after finishing self-driven work on the task. Might LRs and GSs have spent more time earlier in the trial looking at experimenters before switching gaze differentially toward owners? If so, again, we might posit that the CWDs didn't have enough time, post self-driven efforts, to work through the human supplication decision tree.

I don't believe this concern is insurmountable. I do, however, believe the authors need to clearly address possible alternative interpretations to their data given the limitation of time in the current study.

Much more minor concerns:

The general rearing situation of these dogs is similar, which is good. Might it be the case that people interact w/ CWDs differently, even in a home rearing situation? This might at least be addressed. The authors have gone a long way to control for developmental effects, but they have not fully limited all potential avenues.

On line 155, the authors suggest that gazing at the owner might suggest a more "emotional" versus "operational" interest in a human. However, might it not also suggest that the animal has a long prior facultative history with their owner whereby the owner solves lots of problems for the animal? One might just look at one's owner because one's owner usually intervenes in difficult situations, regardless of one's emotional state. In addition, I think it's somewhat loose to suggest that looking at owner then back at task is "asking for help." Certainly we might question whether it bears some similarity to asking for help, but it's much more parsimonious, without specific evidence, to just suggest that the animal is dividing attention between the reward and the means by which prior rewards have often been delivered.

More specific detail on the subjects in the methods/materials would be welcome. Again, I take it the most meaningful contribution here is evidence for genetic, as opposed to developmental, influence on gazing behavior. More detail on life history would be informative.

Line 174 - I did not understand in this training context what was meant by "complete behavioural agreement between experimenters."

Line 334 -- I don't understand this sentence -- how can Marshall-Pescini et al comment on the current (as of yet unpublished) work in an already published work?

385 - The authors suggest that CWD's gazing at experimenters is opportunistic, but might it not also suggest wariness/alertness to a stranger? The familiarization period was short, and in my experience w/ wolves they are VERY attentive to novel humans, in part due to higher arousal/threat sensitivity.

Again, I believe the paper is strong and well written. If the authors can better couch their interpretation in terms of a range of possible interpretations and the study's inherent limitations than I think it certainly merits publication in RSOS.

Reviewer: 2

Comments to the Author(s)
Review on the manuscript RSOS-190946

„WOLF-LIKE OR DOG-LIKE? A COMPARISON OF GAZING BEHAVIOUR ACROSS THREE DOG PURE BREEDS TESTED IN THEIR FAMILIAR ENVIRONMENTS“

written by Maglieri et al.

Overall opinion

This is a very well-written manuscript, scholarly accurate and easy to follow. My main concern is about the (by my opinion) minimal novelty of the conducted research. There is already a number of articles that compared dogs vs wolves, different dog breeds (including Dingoes) gazing behavior in the unsolvable task. This present manuscript adds a new dog breed, the Czechoslovakian Wolfdog to the picture, but the Authors did not provide a good enough reasoning, why did they compare their behavior to the German Shepherd Dogs and Labrador retrievers. If we consider the CWD as a dog-wolf hybrid (still, the extent of wolf alleles should be specified then), it would be a better approach to compare their behavior to socialized wolves and the German Shepherd Dog – because then we would had the results from the two original parent species and their hybrid.

As the German Shepherd Dogs showed intermediate results (not differing from the CWDs neither from the Labrador Retrievers), it is very hard to argue that the CWDs show wolf-like behavior.

Detailed comments

Title – would be more correct probably to write “...across three purebred dog breeds...”

All along the text the correct name is German Shepherd Dog (GSD) and not “German Shepherd”.

Introduction/rationale – the reasoning about the novelty of this research is not convincing enough. As Authors mention, there are published results about the differences among dog breeds' gazing behavior (Jakovcevic et al., Konno et al., Passalacqua et al.), basically reporting that more ancient/ or independently 'working' dog breeds show less human-directed gazing behavior than the cooperative/ or more 'modern' dog breeds. This manuscript is mainly different from only that aspect that Authors tested Czechoslovakian Wolfdogs (CWD) as one of the three tested dog breeds. Although it is known that this dog breed was originally created about 60 years ago by hybridizing wolves and German Shepherd Dogs, Authors did not provide additional information about the still remaining contribution of 'wolf alleles' to the genotype of current CWDs. As from the earlier papers we know how socialized wolves (Miklósi et al., 2003) and dingoes (Smith & Litchfield, 2013) behave in the “gaze during an unsolvable task” paradigm, in case of the present manuscript it is not clear, whether the addition of another dog breed with an unquantified amount of wolf influence would create a significant amount of scientific interest towards the results.

Methods – the comparison between the currently collected and earlier dingo data (retrieved from Smith & Litchfield, 2013) is problematic somewhat, because in the earlier study a very different method was used as 'unsolvable task'.

Results – Lines 276-278: It seems like 0.5 was the expected ratio of subjects that look back at the humans during the solvable task (Binomial test). Why? In case of the solvable task, based on the theory that human-directed gazing is induced by the fact that the dog finds a formerly solvable task as unsolvable, one could expect that no dogs will look back to the human.

Authors should compare of the ratio of dogs gazing at the human and the other gazing-related parameters BETWEEN the two test conditions (solvable vs. unsolvable task), within each dog breeds.

Line 307 – there was no definition for 'gaze alternation' in the Methods. What is the exact description of this behavior?

Discussion

Lines 342-357 – Authors argue that Czechoslovakian Wolfdogs show more wolf-like behavior, because their response in the unsolvable task was different from the retrievers' (less frequent gazing at the humans, more active manipulation of the apparatus). However, it should not be ignored that there was no statistical difference between the performance of CWDs and German Shepherd Dogs in any of the behavioral parameters (as there was no significant difference between the GSDs and the retrievers either). One could say that it is not the CWDs but the retrievers who show the most dog-like behavior... What is with the GSDs? Now are they wolf or dog-like?

Lines 358-361 – Strictly speaking, Binomial test found only a trend ($P=0.054$) in case of German Shepherd Dogs during the unsolvable task, regarding the proportion of dogs that gazed at the human. Therefore

Line 360 – the interpretation of gazing back at the human as “asking for help” does not have a strong basis, it is rather just a commonly (although probably incorrectly) used term.

Line 390 – correctly it should be “Carpathian Wolf” instead of “Carpathian Wolfdog”

Lines 401-403 – the conclusions drawn by the Authors (“In conclusion, despite the numerous crossing with German Shepherds after the first hybrid litter, it seems that the artificial selection operated on Czechoslovakian Wolfdogs has not yet produced a breed showing complete dog-like behavioural features”) is questionable because they did not detect any significant difference between the human-directed gazing behavior of the Czechoslovakian Wolfdog and the German Shepherd Dog.

Author's Response to Decision Letter for (RSOS-190946.R0)

See Appendix A.

Decision letter (RSOS-190946.R1)

08-Aug-2019

Dear Dr Prato-Previde,

I am pleased to inform you that your manuscript entitled "WOLF-LIKE OR DOG-LIKE? A COMPARISON OF GAZING BEHAVIOUR ACROSS THREE DOG BREEDS TESTED IN THEIR FAMILIAR ENVIRONMENTS" is now accepted for publication in Royal Society Open Science.

Kind regards,
Lianne Parkhouse
Editorial Coordinator
Royal Society Open Science
openscience@royalsociety.org

on behalf of Dr Rosalind Arden (Associate Editor) and Kevin Padian (Subject Editor)
openscience@royalsociety.org

Associate Editor Comments to Author (Dr Rosalind Arden):

The Reviewers provided constructive comments on your ms and you have been thorough in taking them into account. Reviewer 1 makes a good point about the difficulty in interpreting whether a dog is expending all its allocated time in trying to figure out a solution to an unsolvable problem, with no time left to 'ask for help', versus a dog that would not arrive at the 'asking for help' option anyway. But I think your response recognises that every study has limitations, you have answered the comments effectively.

I think hybridation is the French form and hybridisation the UK form (line 125).

The sentence below needs to be corrected. Does it refer to the trials of the solvable task?

"As reported in the study by Marshall-Pescini et al. [32], our dogs gazed toward humans to some extent 378 also in the possible trials,"

Aside from these minor points, this manuscript offers an interesting study that makes a very nice contribution to our understanding about the impact of selection on dog behaviour.

Appendix A

Dear Editor,

We would like to thank you for giving us a bit more time to revise the manuscript and the two reviewers for their very useful suggestions. We have carefully considered your suggestions and the reviewers' concerns and comments and we hope that the manuscript in its revised version is now appropriately improved for publication.

Please find below our detailed replies addressing your comments and the comments raised by the reviewers. We have outlined each change made (point by point) as raised in the reviewer comments.

If further changes were necessary, we will be happy to deal with them.

Sincerely yours,

Emanuela Prato-Previde

Response to Editor comments

Editor: *In addition to the Reviewers comments, I should be grateful if you would mention whether or not your results were corrected for multiple testing, and if so how this was done. A sentence or two about any effect sizes would also help the reader to interpret your results.*

Answer: the post-hoc tests we used in the study take into account multiple testing. The effect size was measured using the nonparametric Rosenthal's r coefficient (Field, 2013. Discovering statistics using IBM SPSS statistics. Sage) and has been added in the results section together with p values.

Responses to Reviewers' comments

Reviewer: 1

Rev 1 *I believe the authors' methods were generally sound. I especially commend them for attempting to match their subjects for prior rearing history. Many different breeds are likely to have had very different individual histories with explicit training programs that would be likely to affect problem solving and gazing behavior. Many prior studies have not fully accounted for these potential developmental effects. The paper is well and clearly written and, I think, makes a meaningful contribution to ongoing debate about the genetics underlying dog/human interaction and cooperation.*

Answer: We really thank the reviewer for his/her positive and encouraging comments.

Rev 1 *I have one primary concern with the study and how it was interpreted. Despite some prior work allowing subjects 2 or more minutes for exploring the impossible task, in the current study the subjects had only one minute. I think this limits the confidence with which the data might be interpreted.*

Answer: We agree that in some works the impossible trial lasted up to 2 minutes (usually with wild canids and feral dogs), but in many other studies focused on pet dogs the impossible trial lasted 1 minute. In our study the duration of the impossible trial was the same as in other studies conducted on *problem solving and gazing behaviour* in pet dogs - either mix-breed, pure breed dogs or breed groups (e.g. Marshall-Pescini

et al., 2009; Passalacqua, et al., 2011; Marshall-Pescini et al 2013; Konno, et al.2016; D’Aniello et al., 2015; Scandurra et al., 2015). There is some variability in the studies based on the impossible task paradigm (i.e. the type of apparatus, the number of possible trials and the time allowed to the subjects during the impossible task). Since, in our work comparisons were among dog breeds kept as pets, and not wild canids and feral dogs, we decided to adopt the procedure most often used with dogs. We have briefly explained this point in the text (lines 236-239). In the discussion we have also outlined that “future comparisons using different communicational paradigms as well as problem solving contexts may lead to a clearer interpretation of the current findings” (lines 470-472).

Rev 1. *The authors note that CWDs spend less time looking at humans than do LRs, and suggest this is because LRs are more reliant on humans. However, from figure 4, it is clear that CWDs spent the majority of the minute actively focused on interacting with the apparatus, leaving much less remaining time for looking at humans/asking for/expecting help.*

Answer: This is exactly what we wanted to test, that is the greater independence of the Czechoslovakian Wolfdog that try to solve the task without asking the owner or the experimenter for intervention.

Rev 1. *An alternative interpretation that the authors' data does not clarify is that, instead of CWDs having less interest in humans than LRs, they are just more PERSISTENT in problem solving. It may be that, when one wears out one's self-motivated attempts to solve the problem, each breed might then be equally likely to look to a human for intervention. But it's not clear that CWDs had enough time to meaningfully work beyond their own self-efficacy.*

Answer: CWDs are a recognized dog breed and not wolves, therefore since we wanted to compare their behaviour with other dog breeds we needed to put all the dogs in the same testing condition. We were interested in understanding if CWDs adopt a different strategy compared to other dog breeds. So, the time frame of the task could not be different. We have briefly explained our choice on lines 236-238. We have also outlined that the issue of persistence deserve further attention (lines 400-401).

Rev 1. *I suspect that self-efficacy is indeed anti-correlated w/ communication w/ humans/supplication to/reliance on humans across a range of contexts. These are related phenomena. But I believe these phenomena could still be disentangled with a more careful study design.*

Answer: Thanks for the interesting comment, actually we have outlined the need to address these issues in more detail in the discussion (lines 465-468). We agree that the psychological concept of self-efficacy may well be linked to independent problem solving and that different testing conditions may help in investigating this.

Rev 1. *A similar concern -- the authors note that CWDs spend more time looking at experimenters than owners as opposed to LRs. But, again, I wonder about the time frame/amount of time available for looking after finishing self-driven work on the task. Might LRs and GSs have spent more time earlier in the trial looking at experimenters before switching gaze differentially toward owners? If so, again, we might posit that the CWDs didn't have enough time, post self-driven efforts, to work through the human supplication decision tree.*

Answer: The comparison is within each breed and not across breeds. Even though the CWDs tended to look at humans for a smaller amount of time, they preferentially looked at the experimenter. So, this comparison is independent from the amount of time all the dogs devoted in looking humans.

Rev 1. *I don't believe this concern is insurmountable. I do, however, believe the authors need to clearly address possible alternative interpretations to their data given the limitation of time in the current study.*

Answer: As reported above the time frame/amount of time the dog had during the unsolvable trial was the same of other previous studies, in which dogs could either decide to try individually solve the task or to turn to humans (either the owner or the stranger); in this study all dogs had the same amount of time and used it in a different way. We cannot, of course, exclude that having more time all dogs (and not necessarily only CWDs) would have engaged in different behaviours. We have *address possible alternative interpretations to our data.*

Rev 1- *The general rearing situation of these dogs is similar, which is good. Might it be the case that people interact w/ CWDs differently, even in a home rearing situation? This might at least be addressed. The authors have gone a long way to control for developmental effects, but they have not fully limited all potential avenues.*

Answer: We agree with the reviewer. This is the main problem regarding each study with dogs. It is possible that each owner interacts in a different way with his/her dogs not only at a breed level but also at an individual level. When selecting our sample, we got information about the rearing condition of each dog and we obtained a well-balanced sample with, for example, all the owners sleeping together with the dogs independently from the breed.

We agree with the reviewer that even though the rearing situation of the dogs were as similar as possible it was not possible to control for all environmental variables. Thus, we cannot completely exclude that in everyday life people may interact with CWDs, GSD and GR in a somewhat different way. We have addressed this issue in the discussion (lines 416-420).

Rev 1- *On line 155, the authors suggest that gazing at the owner might suggest a more "emotional" versus "operational" interest in a human. However, might it not also suggest that the animal has a long prior facultative history with their owner whereby the owner solves lots of problems for the animal? One might just look at one's owner because one's owner usually intervenes in difficult situations, regardless of one's emotional state. In addition, I think it's somewhat loose to suggest that looking at owner then back at task is "asking for help." Certainly we might question whether it bears some similarity to asking for help, but it's much more parsimonious, without specific evidence, to just suggest that the animal is dividing attention between the reward and the means by which prior rewards have often been delivered.*

Answer: This is an interesting comment since looking behavior may be influenced also by previous history with the owner. We have changed the term "emotional" into "relational" in the introduction to include prior interactions (lines 175-176). Furthermore, the term "asking for help" has been changed with a more neutral terms throughout the text.

Rev 1- *More specific detail on the subjects in the methods/materials would be welcome. Again, I take it the most meaningful contribution here is evidence for genetic, as opposed to developmental, influence on gazing behavior. More detail on life history would be informative.*

Answer: We added more information in the "Material and Methods" section (lines 191-194).

Rev 1- *Line 174 - I did not understand in this training context what was meant by "complete behavioural agreement between experimenters."*

Answer: The goal of the training was to have experimenters that behaved in a very similar way during testing (same movements and synchronized actions). We agree that the sentence was unclear and we have changed it into: “This aimed at obtaining a high level of behavioural homogeneity between experimenters during the test” (lines 198-199).

Rev 1- Line 334 -- I don't understand this sentence -- how can Marshall-Pescini et al comment on the current (as of yet unpublished) work in an already published work?

Answer: The sentence has been reworded to “As reported in the study by Marshall-Pescini et al. [32]” (line 377).

Rev 1- 385 - The authors suggest that CWD's gazing at experimenters is opportunistic, but might it not also suggest wariness/alertness to a stranger?

The familiarization period was short, and in my experience w/ wolves they are VERY attentive to novel humans, in part due to higher arousal/threat sensitivity.

Answer: This is an interesting point that we have considered (see Discussion Lines 444-448). We can believe that a wild species like wolves can be very attentive to unfamiliar humans, however all the CWDs included in our study were pets and, as we have explained in the subjects section, they were well habituated to interact with different human subjects due to their frequent interactions with strangers.

However, to better investigate this aspect, we compared the dogs' gazing behaviour toward the owner vs the experimenter during the first possible trial. If dogs were alerted by the experimenter they should gaze preferentially at her, but we did not find differences (Results: lines 313-317).

Rev 1- Again, I believe the paper is strong and well written. If the authors can better couch their interpretation in terms of a range of possible interpretations and the study's inherent limitations than I think it certainly merits publication in RSOS.

Answer: The suggestion of the reviewer was taken into consideration and in the discussion some alternative interpretations have been discussed together with potential limitations (Discussion: lines 390-401; lines 416-420; 427-433; line 444-448; lines 463-472).

Reviewer: 2

Rev 2: This is a very well-written manuscript, scholarly accurate and easy to follow.

Answer: We thank the reviewer for giving this very positive opinion

Rev 2: My main concern is about the (by my opinion) minimal novelty of the conducted research. There is already a number of articles that compared dogs vs wolves, different dog breeds (including Dingoes) gazing behavior in the unsolvable task. This present manuscript adds a new dog breed, the Czechoslovakian Wolfdog to the picture, but the Authors did not provide a good enough reasoning, why did they compare their behavior to the German Shepherd Dogs and Labrador retrievers. If we consider the CWD as a dog-wolf hybrid (still, the extent of wolf alleles should be specified then), it would been a better approach to compare their behaviour to socialized wolves and the German Shepherd Dog – because then we would had the results from the two original parent species and their hybrid.

Answer: In the introduction we listed some of the reasons that render the comparison between CWD, GS and LR interesting regardless of previous studies. Probably we have not been sufficiently exhaustive. CWDs are a recent dog breed obtained from an hybridation process and in a limited time frame, whereas the other two breeds are modern breeds resulting from the domestication process followed by a more recent

process of artificial selection. All the adult dogs included in our study shared very similar rearing conditions by having a very strict daily relation with their owners and were pets, living a true pet life. Dingoes and wolves included in previous studies lived in packs and, although socialized with humans, they did not have a real owner, did not live in a human family and consequently they had not developed a strict and continuous relation with humans. A real daily, close, long lasting social interaction with a human owner in a true human social environment is, in our opinion, different from being hand raised and then- at a few months of age- being moved to a farm (Miklosi et al., 2003) or a Sancriuary (Smith and Litchfield, 2013) to live in group ; It seems to us that this could be a sufficiently interesting novelty of our study.

In addition, the few studies based on breed group comparisons in the impossible task provided partially contrasting results (Passalacqua et al., and Konno et al.) and comparison among specific breeds represents an useful and additional tool in understanding the potential genetic influences on modern dogs' communicative abilities, particularly when dogs are exposed to similar rearing and living conditions.

We have now provided further rationale in the Introduction section to better clarify why we chose these breeds to carry out our work (see lines 105-109; 115-125 and 135-146).

Rev 2. *As the German Shepherd Dogs showed intermediate results (not differing from the CWDs neither from the Labrador Retrievers), it is very hard to argue that the CWDs show wolf-like behavior.*

Answer: As we mentioned in the introduction, we expected to find a gradient across these three breeds, with CWDs showing a more to “*wolf-like behaviour*” and the German Shepherd Dog positioned between the other two breeds.

Rev 2: *Title – would be more correct probably to write “...across three purebred dog breeds...”*

Answer: The title has been changed into “ across three different dog breeds...” .

Rev 2: *All along the text the correct name is German Shepherd Dog (GSD) and not “German Shepherd”.*

Answer: We agree that the officially recognized English name of German Shepherds is “German Shepherd Dog”, however more recently the breed’s name was changed back to German Shepherd: indeed, in most studies this breed is mentioned as German Shepherd. We chose a compromise: we have introduced the breed as suggested by the referee and then we moved to the most commonly used name, i.e. German Shepherd along the text (see line 101).

Rev 2: *Introduction/rationale – the reasoning about the novelty of this research is not convincing enough. As Authors mention, there are published results about the differences among dog breeds’ gazing behavior (Jakovcevic et al., Konno et al., Passalacqua et al.), basically reporting that more ancient/ or independently ‘working’ dog breeds show less human-directed gazing behavior than the cooperative/ or more ‘modern’ dog breeds. This manuscript is mainly different from only that aspect that Authors tested Czechoslovakian Wolfdogs (CWD) as one of the three tested dog breeds. Although it is known that this dog breed was originally created about 60 years ago by hybridizing wolves and German Shepherd Dogs, Authors did not provide additional information about the still remaining contribution of ‘wolf alleles’ to the genotype of current CWDs.*

Answer: As explained above, published results about the differences among dog breeds’ in gazing are based on comparisons between breed groups and provide somewhat controversial results (Konno et al., Passalacqua et al.); the work by Jakovcevic et al., is not based on the impossible task but on a different procedure and testing condition.

Czechoslovakian Wolfdogs (CWD) is a really interesting breed because the phase of artificial selection was very fast, compared to the wolf domestication process and it includes direct selection for morphological traits, in contrast with the wolf domestication process, where probably the main selective pressure acted on behavioural traits (Smetanova et al. 2015). Moreover, since CWD is rapidly growing in popularity in different

countries also as a pet - and no scientific studies on this new breed are available so far- it would be interesting to investigate its communicative abilities comparing them with those of two well established breeds.

Additional information about the “still remaining contribution of ‘wolf alleles’ to the genotype of current CWDs” is not available for Italian CWDs dogs, since we do not have clear genetic data at present.

For what concern genetic information available, from the work of Smetanova et al. (2015) on subjects of the same breed from Czech Republik we know that this breed lack of wolf mtDNA haplotypes and that lot of males still carry Y-haplotypes of wolf origin. Lots of subjects still carry wolf microsatellite. Moreover, we know that this breed could have lost or fixed some genes due to the bottleneck and founder effects and genetic drift processes, that are typical processes acting on small population, and to which this breed has undergone. (see Introduction lines 135-137).

Rev 2. *As from the earlier papers we know how socialized wolves (Miklósi et al., 2003) and dingoes (Smith & Litchfield, 2013) behave in the “gaze during an unsolvable task” paradigm, in case of the present manuscript it is not clear, whether the addition of another dog breed with an unquantified amount of wolf influence would create a significant amount of scientific interest towards the results.*

Answer: In the works by Miklosi et al on wolves and Smith & Litchfield on dingoes the subjects were only a few months old and were hand reared. They were wild canids habituated to human presence, but they were not “pets”. As already reported above CWDs are dogs, with a real daily, close, long lasting social interaction with a human owner in a true human social environment: this is in our opinion quite different from being hand raised and then- at a few months of age- being moved to a farm (Miklosi et al., 2003) or a Sanctuary (Smith and Litchfield, 2013) to live in group.

Rev 2. *Methods – the comparison between the currently collected and earlier dingo data (retrieved from Smith & Litchfield, 2013) is problematic somewhat, because in the earlier study a very different method was used as ‘unsolvable task’.*

Answer: The apparatus used in the work of Smith & Litchfield on dingoes was different but based on a similar concept: allowing the animals to experience that the problem was solvable and make the problem unsolvable. Dingoes are wild dogs, and the authors used the same apparatus used by Miklosi with wolves, to compare these wild canids. Given the goal of the task was the same, we thought that the comparison would be interesting. We have now explained this in the text (lines 272-277).

Rev 2. *Results – Lines 276-278: It seems like 0.5 was the expected ratio of subjects that look back at the humans during the solvable task (Binomial test). Why? In case of the solvable task, based on the theory that human-directed gazing is induced by the fact that the dog finds a formerly solvable task as unsolvable, one could expect that no dogs will look back to the human.*

Answer: During the solvable task, we assumed that the behaviour was randomly distributed (0.5). Not all dogs immediately understand the task so, before reaching the food, they may gaze at humans. Initially, the situation is new and ambiguous for every dog, so gazing at humans is not so surprising.

Rev 2- *Authors should compare of the ratio of dogs gazing at the human and the other gazing-related parameters BETWEEN the two test conditions (solvable vs. unsolvable task), within each dog breeds.*

Answer: In previous studies (both on wild canids and dogs) using the impossible task paradigm this type of comparison is almost absent since the possible trials are considered just a form of pre-training allowing subjects to experience that the task is solvable before presenting them with the impossible task (Miklosi et al., Passalacqua et al., Konno et al, etc). Indeed, dogs were admitted to the impossible task only if they

succeeded in retrieving the food in 4 out of 6 possible trials. Moreover, the duration of the possible trials is very variable across trials due to dogs learning to solve the task as the number of trials increases (from 1 to 6) and due to the individual variability of the dogs (some dogs obtained the reward in 3 seconds and some other in 60 seconds). For these reasons making comparison is difficult and possibly questionable.

Rev 2- Line 307 – *there was no definition for ‘gaze alternation’ in the Methods. What is the exact description of this behavior?*

Answer: Thanks. We have added the definition of gaze alternation in the Methods (lines 260-261).

Rev 2- Discussion

Lines 342-357 – Authors argue that Czechoslovakian Wolfdogs show more wolf-like behavior, because their response in the unsolvable task was different from the retrievers’ (less frequent gazing at the humans, more active manipulation of the apparatus). However, it should not be ignored that there was no statistical difference between the performance of CWDs and German Shepherd Dogs in any of the behavioral parameters (as there was no significant difference between the GSDs and the retrievers either). One could say that it is not the CWDs but the retrievers who show the most dog-like behavior... What is with the GSDs? Now are they wolf or dog-like?

Answer: Yes, Labrador Retrievers seems to be the most human-oriented breed of our sample. This is probably linked to their higher sociability; retrievers’ sociability towards people has been reported in studies directly comparing GS and LR behaviour (e.g. Serpell et al. 2014; Jakovcevic et al. 2010). We have addressed this in the discussion (430-433). In addition, CWDs are at the opposite side and thus in the discussion we have reported that they are behaviourally more similar to ancient breeds (463-465).

Rev 2. Lines 358-361 – *Strictly speaking, Binomial test found only a trend (P=0.054) in case of German Shepherd Dogs during the unsolvable task, regarding the proportion of dogs that gazed at the human.*

Answer: Yes we agree it is a strong trend.

Rev 2. Line 360 – *the interpretation of gazing back at the human as “asking for help” does not have a strong basis, it is rather just a commonly (although probably incorrectly) used term.*

Answer: This is true, even if the term “asking/seeking for help” is widely used in the literature. We have modified it using more neutral term throughout the text.

Rev 2- Line 390 – *correctly it should be “Carpathian Wolf” instead of “Carpathian Wolfdog”*

Answer: Done

Rev 2- Lines 401-403 – *the conclusions drawn by the Authors (“In conclusion, despite the numerous crossing with German Shepherds after the first hybrid litter, it seems that the artificial selection operated on Czechoslovakian Wolfdogs has not yet produced a breed showing complete dog-like behavioural features”) is questionable because they did not detect any significant difference between the human-directed gazing behavior of the Czechoslovakian Wolfdog and the German Shepherd Dog.*

Answer: Yes, it is true. We have softened our conclusions taking into account this observation (see lines 462-465).